# Assessing the Potential of Small Peptides for Altering Expression Levels of the Iron-Regulatory Genes *FTH1* and *TFRC* and Enhancing Androgen Receptor Inhibitor Activity in In Vitro Prostate Cancer Models

**DOI:** 10.3390/ijms242015231

**Published:** 2023-10-16

**Authors:** Crawford Currie, Christian Bjerknes, Tor Åge Myklebust, Bomi Framroze

**Affiliations:** 1HBC Immunology Inc., 1455 Adams Drive, Suite, Menlo Park, CA 2043, USA; bf@hbcimmunology.com; 2Hofseth Biocare, Keiser Wilhelmsgate 24, 6003 Ålesund, Norway; chbj@hofsethbiocare.no; 3Department for Health Sciences, Faculty of Medicine and Health Sciences, Norwegian University of Science and Technology, 6025 Ålesund, Norway; 4Department of Registration, Cancer Registry of Norway, 0379 Oslo, Norway; tor.age.myklebust@helse-mr.no; 5Department of Research and Innovation, Møre og Romsdal Hospital Trust, 6026 Ålesund, Norway; 6GPH Biotech LLC, 1455 Adams Drive, Menlo Park, CA 94025, USA

**Keywords:** anticancer peptides, prostate cancer, iron metabolism, iron regulating peptides, bioactive oligopeptides, chemosensitization, *FTH1* modulation

## Abstract

Recent research highlights the key role of iron dyshomeostasis in the pathogenesis of prostate cancer (PCa). PCa cells are heavily dependent on bioavailable iron, which frequently results in the reprogramming of iron uptake and storage pathways. Although advanced-stage PCa is currently incurable, bioactive peptides capable of modulating key iron-regulatory genes may constitute a means of exploiting a metabolic adaptation necessary for tumor growth. Recent annual increases in PCa incidence have been reported, highlighting the urgent need for novel treatments. We examined the ability of LNCaP, PC3, VCaP, and VCaP-EnzR cells to form colonies in the presence of androgen receptor inhibitors (ARI) and a series of iron-gene modulating oligopeptides (FT-001-FT-008). The viability of colonies following treatment was determined with clonogenic assays, and the expression levels of *FTH1* (ferritin heavy chain 1) and *TFRC* (transferrin receptor) were determined with quantitative polymerase chain reaction (PCR). Peptides and ARIs combined significantly reduced PCa cell growth across all phenotypes, of which two peptides were the most effective. Colony growth suppression generally correlated with the magnitude of concurrent increases in *FTH1* and decreases in *TFRC* expression for all cells. The results of this study provide preliminary insight into a novel approach at targeting iron dysmetabolism and sensitizing PCa cells to established cancer treatments.

## 1. Introduction

Patients with advanced-stage prostate cancer (PCa) represent a group with significant unmet medical needs in terms of available treatment options and clinical outcomes. Among middle-aged men living in developed countries, PCa is more prevalent than any other form of cancer [1]. As a highly heterogeneous disease, PCa encompasses a spectrum of tumors with varying clinical outcomes. In about 80% of initially diagnosed cases, the prostate tumor is confined locally to the prostate gland [2]. Locally advanced disease accounts for approximately 15% of cases. Another 5% represents the cases presenting with distant metastasis. Surgery and radiotherapy, with or without concurrent androgen-deprivation therapy (ADT), can be employed with curative intent in the case of localized PCa [3]. In such cases, the outlook is generally good, with a 97% survival rate given a moderately low recurrence rate. [1]. Metastatic PCa, however, has a dismal 5-year survival rate of 30% [4]. Clinical responses to second-generation androgen receptor inhibitors (ARIs), primarily enzalutamide, are often inconsistent and unpredictable, with only transiently beneficial effects. Advanced-stage disease is characterized by disease heterogeneity, which makes one-size-fits-all treatment generally ineffective. Moreover, drug toxicity may limit the dose necessary to achieve an adequate therapeutic response. While antiandrogen therapies have made significant progress, PCa diagnosed at an advanced stage is almost always fatal. It is evident that novel treatment strategies are required, as evidenced by a 3% annual increase in incidence of PCa, 50% of which is due to advanced-stage disease [5]. Developing antiandrogens that are both safe and effective at advanced stages of the disease remains a major challenge in the present day. New advances in cancer research have shed light on one aspect of tumor biology that can be targeted in order to enhance the effectiveness of existing treatments. Iron dysmetabolism is a metabolic adaptation that occurs frequently in aggressive and undifferentiated tumors, and it may be exploited to sensitize iron-addicted PCa tumors to standard cancer therapy. In several studies, it has been demonstrated that aggressive tumors have a larger labile iron pool (LIP), the intracellular aggregate of bioavailable iron, than those of less aggressive tumors. In order to satisfy the high growth potential of highly malignant cancer cells, sufficient amounts of bioavailable iron are required for mitochondrial respiration and DNA replication [6,7]. Therapeutic approaches that exploit this metabolic adaptation, which is a particularly common molecular event in advanced PCa, may be highly relevant in order to improve current treatments and circumvent disease heterogeneity. In malignant cells, influencing iron homeostasis likely has effects on cell viability that go beyond affecting mitochondrial respiration and DNA replication. Several important pathways intersect with the iron metabolism axis, including those governing immunity, inflammation, and androgen receptor (AR) signaling [8]. Importantly, the relative abundance of the intracellular LIP may affect the expression of genes that are tumor suppressive. Indeed, there has been a considerable amount of research conducted that indicates that iron deprivation activates specific tumor-suppressive functions, the outcome of which may sensitize the PCa cell to cancer therapy and constitute a partial phenotypic rescue. Various exploitation strategies of iron metabolism have been considered in previous studies, including iron overload of PCa cells [9]. The result is ferroptosis, a form of apoptosis that has been recognized in recent years. The use of short anticancer peptides derived from natural sources may be another strategy for targeting iron metabolism at various levels. Natural lead structures have the advantage of having a higher safety profile. Bioactive peptides are recognized broadly to be intrinsically target-specific compounds that are generally regarded as safe [10]. The properties and versatility of these peptides can be further improved by applying a variety of chemical modifications.

Testosterone and dihydrotestosterone are two of the principal androgens identified as contributing to PCa progression. At the same time, a normal and disease-free prostate gland requires androgens for normal development and function. As androgens cross the cell membrane, the molecules form a complex with the cytosolic AR, which triggers a cascade of events in which nuclear translocation occurs. Modulation of gene expression occurs in the nucleus, generally in genes involved in maintaining, growing, and differentiating the cell [11].

While androgens are necessary for prostate homeostasis under physiological conditions, AR signaling is a common central driving factor in the pathogenesis of advanced-stage PCa. The term castration-resistant prostate cancer (CRPC), sometimes called hormone-resistant prostate cancer (HRPC), provides an accurate terminology of the category referred to as advanced-stage PCa. CRPC may be metastatic (mCRPC) or non-metastatic. A castration-resistant condition is characterized by an innate resistance to hormone manipulation. At many disease stages, however, dependency on AR signaling is retained. Among the treatment methods that exploit this dependency is ADT, which has been regarded as a cornerstone of PCa treatment since 1996 [12]. Pharmacological androgen deprivation encompasses varying modalities beyond the scope of this article but includes luteinizing hormone-releasing hormone agonists (LHRH agonists), LHRH antagonists, CYP17 inhibition, and antiandrogen therapy [13]. ADT is initially effective in the treatment of PCa, but ultimately leads to the development of CRPC as a clinical outcome [14]. Mutations involving the AR is often an important step in the transformation to CRPC phenotypes, which may involve constitutively active AR splice variants, which alter the cell sensitivity to hormone stimulation [15]. Once CRPC transformation has occurred, innate biological complexity makes this a particularly problematic group of tumors that respond poorly to further hormonal manipulation. In some patients with CRPC, the tumor lesions exhibit a particularly aggressive form of disease, with low to no AR expression (AR-null), often showing undifferentiated neuroendocrine features [16]. ADT as a treatment modality is complicated by observations that suggest that a number of sources are capable of de novo androgen synthesis. An example of this is the synthesis of androgens by the intestinal microbiome, which may compromise the therapeutic effectiveness of ADT modalities [11,17,18,19]. Although incurable, mCRPC, the most malignant disease state, can still be treated, with the intent of life extension [20]. An important treatment modality for mCRPC is the second-generation antiandrogen enzalutamide, which inhibits androgen receptors to prevent androgen signaling [21]. While enzalutamide confers beneficial treatment outcomes, lineage plasticity may often result in enzalutamide resistance. This can be observed approximately six to twelve months after treatment has been initiated [22,23]. Additionally, bicalutamide, a first-generation antiandrogen drug, is approved for use in treating advanced-stage prostate cancer in combination with LHRH analogs [24]. Aside from exerting a stronger AR blockade, enzalutamide inhibits nuclear translocation and has demonstrated superior clinical outcomes in patients with CRPC [25]. Multimodal treatment procedures, or treatments applied in more than one layer, have therefore become increasingly relevant for achieving improved disease control and patient outcomes. Treatments of this type may be more effective at addressing disease heterogeneity, as well as maintaining a low level of toxicity [26,27]. It is possible that combinations can improve side effect profiles by reducing the dosage required for individual drugs to achieve the desired therapeutic response. Efforts are continually being made to improve the current therapeutic landscape and to improve imaging technology in order to facilitate a more accurate diagnosis and follow-up as well as to identify occult lesions, thereby advancing personalized prostate cancer treatments [28,29,30,31,32,33,34,35,36,37,38]. However, despite ongoing research efforts and two decades of declining mortality, recent cancer statistics indicate an alarming 3% annual rise in PCa incidence from 2014 to 2019 [1].

Bioactive peptides of marine origin are widely recognized for having bioactivities relevant to a number of human diseases [39]. It is not uncommon for marine protein hydrolysates to contain anticancer peptides encrypted within their parent proteins [40]. Several marine-derived compounds have been reported to exhibit anticancer properties in PCa models [41,42]. While there is a potentially wide range of mechanisms through which their anticancer effects may be mediated, in pre-clinical studies, these peptides have been shown to sensitize cancer cells to anticancer agents [43,44]. While some challenges remain before marine-based peptides may have widespread clinical applications, some anticancer peptides have reached the clinical testing stage [45].

Previous preliminary studies by the authors revealed that a soluble protein hydrolysate (SPH) was capable of significantly altering the expression of ferritin heavy chain 1 (*FTH1*) and transferrin receptor (*TFRC*) expression in gingival epithelial cells [46]. In a follow-up study, this SPH was shown to significantly enhance the efficacy of a first-generation antiandrogen in two types of PCa cells [47]. The extent to which gene modulation was altered corresponded to the magnitude of the effect on suppressing growth of PCa cells in the same study. In the current study, four PCa cell models that show distinct phenotypic characteristics ranging from mild to highly aggressive are employed in order to validate the previous findings in vitro. The primary sequence of the peptide responsible for modulating *FTH1* and *TFRC* was identified, and a total of eight bioactive peptides were designed around a common core sequence for application in clonogenic and gene expression assays. For the purpose of capturing disease heterogeneity, LNCaP, PC3, VCaP, and VCaP-EnzR cells were chosen for the assays.

The excessive reliance on bioavailable iron during PCa tumor development can negatively impact growth due to oxidative stress caused by too much iron, or the deleterious effects of iron deficiency. A delicate balance must be maintained by the malignant cells in order to maintain an appropriately dysregulated iron metabolism, as too much or too little iron can have adverse effects on cell viability. *FTH1* expression is a key iron-regulatory gene that impacts cytosolic LIP, which if significantly perturbed may affect cancer growth [48,49,50]. Indeed, access to bioavailable iron can be directly altered through modulating the expression of both *FTH1* and *TFRC*. In addition, *FTH1* expression is associated with antioxidant activity, angiogenesis regulation, and direct interaction with other signaling pathways along its expression axis that has implications for cell phenotype [51,52,53,54]. It is noteworthy that *FTH1* expression may have innate antiproliferative properties in PCa through a mechanism of sequestering oncogenic microRNA (miRNA) through common response elements on *FTH1* pseudogene transcripts [55]. Due to the intersection of iron metabolism and tumor-suppressive pathways, it is likely that tumor-suppressive pathways are affected in conditions of altered expression of *FTH1* [56].

Ultimately, the study serves as preliminary research in a drug discovery and development process of therapeutic peptides. Further in vitro and preclinical studies will be conducted on the most potent and promising lead peptides in order to continue the development of peptide compounds that may have clinical applications in the context of PCa treatment.

## 2. Results

### 2.1. VCaP Clonogenic Survival Following Treatment with SPH and Enzalutamide

Figure 1 shows the relative % colony survival rate for VCaP cells under various treatment conditions. The plating efficiency, defined as the proportion of inoculated cells that develop into colonies, is expressed as a relative % colony survival rate compared with an untreated cell population where plating efficiency is arbitrarily set to 100. Enzalutamide treatment at the highest assay dose (10.0 µM) resulted in significant growth inhibition of VCaP cell colonies (*p* < 0.01) relative to DMSO control (Figure 1). In the presence of 160 µg/mL SPH, enzalutamide at 1.4 µM (half-maximal inhibitory concentration; IC_50_) significantly suppressed (*p* < 0.05) the clonogenic potential of VCaP cells. SPH alone did not significantly affect the clonogenic potential of VCaP cells at any of the tested concentrations. However, a modest non-statistical numerical reduction in colonies may be observed. There was no statistically significant growth inhibition of VCaP cells with 1.0 µM enzalutamide alone. Simultaneous application of 160 µg/mL SPH in combination with 1.4 µM enzalutamide achieved statistical growth inhibition at a concentration of enzalutamide multiple-fold lower than what was required to achieve statistical growth inhibition with enzalutamide treatment alone. This appears to be a potentiation of antiandrogen activity and may be indicative of a synergistic effect. For the combination effects to be statistically significant, a threshold concentration of SPH appears to be necessary.

### 2.2. VCaP-EnzR Clonogenic Survival Following Treatment with SPH and Enzalutamide

VCaP-EnzR cells harbor acquired enzalutamide resistance, commonly mediated through the expression of AR splice variants. In line with expectations, growth suppression did not occur at most tested enzalutamide concentrations. However, significant colony suppression (*p* < 0.05) was evident at very high enzalutamide concentrations (100.0 µM; Figure 2). At very high concentrations, growth inhibitory effects may be attributable to non-specified secondary off-target effects not otherwise present at lower doses. The highest and most significant anti-proliferative effect was observed with enzalutamide at a corresponding IC_50_ value (47 µM) when co-administered with 160 µg/mL SPH (*p* < 0.01), a finding that may be considered to indicate synergy.

### 2.3. LNCaP Clonogenic Survival Following Peptide and Bicalutamide Co-Treatment

Peptides FT-002 (10 µM) and FT-005 (10 µM) exhibited the greatest anti-proliferative effects in LNCaP cells in combination with 0.4 µM bicalutamide (IC_50_; Figure 3). The viability of the LNCaP colony survival rate was reduced from 50% seen with bicalutamide alone (IC_50_) to 9% with concurrent administration of FT-002 and bicalutamide (*p* < 0.001). The magnitude of growth inhibition of FT-005 and bicalutamide were comparable to that of FT-002 and bicalutamide. However, a final colony viability of 14% was a somewhat less pronounced effect (*p* < 0.05). While bicalutamide’s suppressive effects on colony growth was enhanced by the addition of 160 µg/mL SPH, the difference in growth suppression compared to bicalutamide treatment alone (IC_50_) was not statistically significant.

### 2.4. PC3 Clonogenic Survival Following Peptide and Bicalutamide Co-Treatment

Among all tested peptides, peptides FT-002 and FT-005 in combination with bicalutamide exhibited the strongest anti-proliferative effects on PC3 colony formation potential (Figure 4). Peptide FT-005 co-administered with bicalutamide (IC_50_) suppressed PC3 colony formation (11% viability, *p* < 0.01) more effectively than FT-002 (16% viability, *p* < 0.05), which is contrary to the results obtained for the LNCaP cell assay. All compounds at their tested concentrations showed an overall less pronounced suppression of PC3 colony formation potential than what was observed for LNCaP cells, highlighting the comparably indolent nature of the LNCaP cell line. The suppression was, however, still significant compared to the treatment with bicalutamide alone (IC_50_). This is an intriguing set of results, as PC3 cells are regarded as an aggressive and undifferentiated cell line and in vitro model of advanced PCa. A high growth potential is typically associated with aggressive phenotypes [57]. Despite these characteristics, peptide and bicalutamide co-treatment resulted in significant suppression of colony growth, proposing a mechanism that seems applicable across phenotypes. The lack of functional signaling typical of PC3 cells is a characteristic that requires high levels of mitochondrial respiration, which may help explain this phenomenon [9]. In order to maintain high mitochondrial respiration, iron is required in the form of iron–sulfur clusters and heme that are incorporated into various mitochondrial proteins [58]. This metabolic state may be particularly sensitive to perturbations affecting availability to iron.

### 2.5. VCaP Clonogenic Survival Following Peptide and Enzalutamide Co-Treatment

There was a similar pattern of growth suppression observed in this assay as observed in the PC3 assay. As a result of concurrent treatment with FT-005 and enzalutamide (FT-005-E), the colony formation potential was significantly diminished (*p* < 0.01), decreasing from 50% for the control condition (1.4 µM enzalutamide; IC_50_) to 21% for FT-005-E (Figure 5). A significant growth suppression was also observed with FT-002-E, resulting in a final 23% colony survival rate (*p* < 0.05).

### 2.6. VCaP-EnzR Clonogenic Survival Following Peptide and Enzalutamide Co-Treatment

A significant decrease in colony survival was observed for VCaP-EnzR when incubated with FT-002-E (*p* < 0.01) and FT-005-E (*p* < 0.05) (Figure 6). Despite the resistance of these cells to enzalutamide, sensitivity to enzalutamide was retained at a corresponding IC_50_ value of 47 mM. Notably, the growth suppression pattern observed in VCaP-EnzR cells was more pronounced as a result of the peptide and antiandrogen treatments than what was observed in VCaP cells. The concentrations of enzalutamide employed for this cell line were, however, necessarily higher due to innate resistance, preventing a direct comparison. Double combination treatment (FT-002 + IC_50_ enzalutamide) resulted in a cell viability of only 11%, corresponding to an approximately four-fold reduction when compared to treatment with enzalutamide (IC_50_) alone.

### 2.7. Gene Expression Analysis

All four cell lines were examined for differential *FTH* and *TFRC* expression under different treatment conditions. As a threshold for identifying significant gene expression changes, an average fold change threshold of two was used. We further examined the differential gene expression of the four most potent peptides with regard to their anti-proliferative properties.

#### 2.7.1. VCaP and VCaP-EnzR Transcriptional Alterations Following SPH and Enzalutamide Treatment

A mean two-fold upregulation of *FTH1* mRNA levels (Figure 7) was observed for cell lines VCaP and VCaP-EnzR after incubation with SPH at 160 µg/mL and enzalutamide at concentrations of 1.4 µM and 47 µM, respectively.Both cell lines showed transcriptional downregulation of *TFRC* following combination treatment (Figure 8), although the reduction did not reach the significance threshold of a two-fold reduction.

#### 2.7.2. LNCaP and PC3 Transcriptional Alterations Following Peptide and Bicalutamide Treatment

With the exception of simultaneous FT-006 and bicalutamide treatment, all peptide–antiandrogen combinations significantly increased the expression of *FTH1* mRNA in LNCaP cells (Figure 9). *FTH1* expression was significantly increased in PC3 cells by peptides FT-005 and FT-002 in combination with bicalutamide. In neither cell line was *TFRC* significantly downregulated (Figure 10), but robust numerical changes in expression were evident, particularly for FT-005-B.

#### 2.7.3. VCaP and VCaP-EnzR Transcriptional Alterations Following Peptide and Enzalutamide Treatment

VCaP and VCaP-EnzR showed marked differences in responses to peptide and enzalutamide treatments. As shown in Figure 11, significant upregulation of *FTH1* was observed for all peptide combinations in both cell types, with the exception of FT-007-E in VCaP cell lines. The transcriptional effects were more pronounced for VCaP-EnzR cells, particularly for FT-002-E, where the expression of *FTH1* mRNA increased by more than 3.5-fold. In both cell lines, the majority of the peptides tested decreased *TFRC* expression close to the threshold for significance. In VCaP-EnzR cells, only FT-002-E and FT-005-E demonstrated statistically significant downregulation of *TFRC* according to our defined criteria (Figure 12). This difference in results could be attributed to their differing phenotypes with respect to androgen signaling and corresponding iron requirements. VCaP cell lines are recognized to express AR at high levels [59,60]. According to some reports, however, AR signaling in VCaP-EnzR cells appears to be even greater than that observed for VCaP cells [61]. Consequently, cells with a VCaP-EnzR phenotype may be more adversely affected than those with a VCaP phenotype when iron metabolism is shifted toward a pro-storage iron phenotype, which is facilitated by the increased translation of high amounts of *FTH1* mRNA transcripts to yield ferritin heavy chains. It seems reasonable to assume that increased availability of ferritin heavy chains would stimulate the production of ferritin, to store iron in its inactive form.

## 3. Discussion

While there has been a reduction in overall cancer mortality by 33% since 1991 [1], prostate cancer incidence is on the rise. Novel treatment approaches are necessary to address this disease and, in particular, approaches that can account for the complex heterogeneous biology characteristic for more advanced-stage PCa disease.

In this study, we investigated the combined antiproliferative effects of bioactive peptides and selected ARIs in in vitro PCa models. Our study evaluated four cell lines as an approximation of PCa disease heterogeneity. An assessment of the colony growth potential following various treatments and their corresponding gene expression levels with respect to two key iron-regulating genes was then conducted. Across all four tested cell lines, bicalutamide and enzalutamide were more effective at inhibiting tumor growth when administered in combination with any of the eight peptides. Consistently, however, FT-002 (REESGEP; 0.8028 kDa) and FT-005 (LDEESGEP; 0.8748 kDa) exerted the greatest antiproliferative activity on the cell line colonies when administered in combination with antiandrogen compounds. PC3 and VCaP cells showed the greatest response to FT-005-B/-E, with a five-fold and two-fold reduction in clonogenic potential, respectively. Combined application of peptide FT-002 and bicalutamide resulted in a six-fold reduction in survival of LNCaP colonies. VCaP-EnzR cells responded similarly to the peptide combinations FT-002-E and FT-005, with colony-forming capacities decreasing approximately five-fold. Interestingly, a consistent finding across all tested cells is the observation that colony growth suppression generally corresponds to the degree of increase in *FTH1* and a decrease in *TFRC* mRNA levels. The magnitude of this gene expression signature did not, however, invariably and directly correlate with colony suppression, as highlighted by the observation that FT-007-B-exposed PC3 colonies demonstrated expression patterns similar to that of FT-005-B-exposed PC colonies, with colony suppression still differing by two-fold. It should be noted that suppression was observed across all cell lines treated, suggesting that the mechanism may be applicable to a wide range of malignant PCa cells. The validity and clinical applicability of this mechanism in addressing the heterogeneity of PCa remains to be determined. The co-treatment also inhibited the growth of the AR-null PC3 cells, which are phenotypically highly aggressive PCa cells [62]. The mean colony survival rate was reduced to a mere 11% with peptide–antiandrogen applications for PC3 colonies. Lastly, after incubation with FT-005-E, the colony survival rate of VCaP cell colonies decreased to 23%. We may summarize this by stating that a general agreement appears to exist between the data and results presented herein and those reported in our previous in vitro study employing an SPH and bicalutamide treatment protocol in PCa cells [47].

The exact mechanisms responsible for peptide-mediated growth suppression and *FTH1* and *TFRC* modulation require further investigation. By investigating the amount of free iron within the cells, as well as relevant genes upstream and downstream of the genes assessed in this study, it seems possible to gain a deeper understanding of the mechanisms involved. It is certainly scientifically plausible that perturbations of iron-regulatory mechanisms will have significant implications in terms of cell viability and growth when considering the central role of a properly regulated iron homeostasis in cellular respiration and DNA replication. Indeed, it is recognized that altered expression patterns of genes related to iron metabolism have prognostic significance in the case of PCa [63,64]. Observations of chemosensitization of PCa cells in vitro following intracellular iron depletion, which has been reported in past research, may be attributed to a complex interaction between iron metabolism pathways and pathways involved in inflammation signaling, immunity, AR signaling, and tumor suppression, perhaps specifically those mediated through N-myc downstream regulated gene 1 (*NDRG1*) [65,66]. The function of *NDRG1* in the context of PCa appears to be related to cell growth, iron regulation, and androgen signaling [67], which may be hypothesized to be one of probably several important mechanisms mediating growth suppression of PCa cell lines. A possible mechanism of peptide modulation of iron-regulatory gene expression may alter the activity of iron-responsive element-binding protein 2 (IRP2), which regulates the stability and translational activity of these genes through gene transcripts [68,69]. Earlier studies of IRP2 knockdown in PCa cells have demonstrated similar gene expression patterns to those observed in this study, which may provide some support for this hypothesis [69].

There is evidence that upon transcription of *FTH1*, at least in the case of the PC3 and DU145 cell lines, a set of genes originating from the parental transcripts are produced in equimolar amounts to *FTH1* transcripts which are referred to as *FTH1* pseudogenes [64]. *FTH1* pseudogenes are long non-coding RNAs which are generated from retrotransposition of their parent transcripts [55]. In the context of oncogenesis, and specifically tumor suppression, several pseudogenes belonging to the ferritin family of genes seem to exert an important regulatory role. Recent research indicates that the *FTH1* pseudogenes play a key role in tumor suppression. In PCa, an extensive network of oncogenic microRNAs (miRNAs) regulates the expression of genes that affect tumorigenesis and chemosensitivity [70,71]. While *FTH1* are frequent targets of such oncogenic miRNAs, *FTH1* pseudogenes display sequence homology to their parent transcript, which allows pseudogenes to compete for binding to oncogenic miRNAs through their common response elements [72,73]. Pseudogene “sponging” of various oncogenic miRNAs alters their relative intracellular levels, the balance of which may release their downstream gene targets from suppression. Genes suppressed by oncogenic miRNAs are genes commonly responsible for inducing tumor suppressive functions. The significance of *FTH1* pseudogenes in controlling gene expression in PCa is also evidenced by a study showing that *FTH1* knockdown diminishes levels of p53, a known oncogenic miR-638 target and significant phenotypic rescue protein [64]. In light of *FTH1*’s apparent involvement in oncogenesis, its expression in PCa may be of greater significance than currently realized in terms of its influence on tumor suppression. A further investigation of the miRNA network seems warranted in light of this somewhat novel understanding of cellular miRNA networks.

In addition to its wide electron-transfer capabilities, cellular iron plays a crucial role in mitochondrial energy production. Moreover, iron is an essential co-factor for functional DNA replication enzymes. Clinical trials employing iron chelator treatment modalities, however, have not found benefits for patients harboring CRPC. Due to the non-specificity of iron chelation, its use in clinical settings has been questioned. Studies have shown that forced expression of ferroportin in PCa cells reduces growth and depletes LIP, a mechanism that is analogous to the gene modulation observed in this study. The investigation of the status of the LIP seems to be an essential step in advancing current peptide research. Intriguingly, the state of intracellular iron depletion constitutes a signal to engage tumor suppressive pathways in cancer cells, specifically *NDRG1* expression being a common occurrence. The recognition that *NDRG1* links iron metabolism and metastatic potential is another intriguing fact that warrants consideration for future research. While having pleiotropic properties, in the context of PCa, *NDRG1* functions as broadly tumor-suppressive. Due to its function as a gene regulated by iron that also interacts with the androgen signaling axis, significant research attention has been given to its potential role in phenotypic rescue of cancer cells and tumor suppression.

Certain peptides may be able to modulate gene expression through some of their physicochemical characteristics. Short peptides may behave as epigenetic modulators by modifying histones or the DNA methylation status of specific gene promoters. Short peptides of 2–7 amino acids may be capable of penetrating the cell nucleus and interacting with histones to separate double-stranded DNA in order to enhance transcriptional availability of certain gene promoters [74]. Furthermore, as peptides with glutamate-enriched backbones, DNA unwinding properties would be expected [75]. *FTH1* and *TFRC* are genes that both have CpG islands in their promoter regions, whose methylation status determines their expression levels [76]. Peptide-mediated methylation of such CpG islands may be achieved through modulation of DNA methyltransferases and protein arginine methyltransferases. Whether these mechanisms mediate gene expression in the case of the peptides investigated in this study is another question that remains to be answered.

We may conclude, based on the data presented in this study and the supporting research presented, that peptides containing the specific core amino acid sequence EESGE coordinately increase the expression of *FTH1* and decrease the expression of *TFRC* in PCa cells. We propose that this reduces the amount of free iron available to cells, thereby disrupting centrally important metabolic pathways, which prevents the malignant cell from buffering the antiproliferative pressure caused by the presence of antiandrogens. As an extension of this hypothesis, it might be interesting to consider whether the LIP engages *NDRG1* expression to play a primary role in this chemosensitization process [77,78]. A future study should include the likely involvement of various proliferative pathways, tumor-suppressive genes, and the LIP. As research has shown that *FTH1* mediates a tumor suppressor role in PCa, it would be prudent at some point to examine the pseudogene-miRNA network. Some conflicting in vitro results exist. According to some in vitro studies, *FTH1* knockdown impaired PC3 cells’ migratory abilities, which are outcomes that would be unfavorable in a clinical setting [50]. The significance of these findings is unclear, and more research needs to be conducted in order to determine the implications of these findings.

There are some limitations to this study that should be noted. Formally assessing *FTH1* and *TFRC* mRNA expression relative to a fold change threshold may be considered a study limitation, as noisy data may contribute to false positives in triplicate experiments with lower statistical power. Another consideration is the phenomenon of cellular cooperation in which proliferation rates may vary among cell lines due to auto-/paracrine growth mechanisms not accounted for in the assay, thus possibly skewing survival results [79]. Growth rates and densities, however, were consistent across controls, which suggests that cellular cooperation is low and may have occurred fortuitously only at seeded densities.

Considering the wide range of molecular mechanisms that contribute to heterogeneity and complexity of prostate cancer, it may be difficult to imagine a universal cure. Personalized PCa treatment could be offered through the augmentation of currently available therapies that in isolation often produce inadequate clinical outcomes. Employing specific peptides that perturb cancer cell iron metabolism across histopathological cancer subtypes may be one such avenue. The simultaneous alteration of *FTH1* and *TFRC* expression may be capable of mimicking the effects of iron chelation and its resulting intracellular iron depletion [80], while avoiding their many non-specific and undesirable side effects. Since iron metabolism intersects with a number of important molecular pathways that are associated with drug resistance, this could offer a strategy to overcome ARI resistance. Peptides may be designed around activities with desirable biochemical and anticancer activities. Consequently, treatments may be increasingly tailored on a case-by-case basis. Future studies need to delineate the exact mechanisms underlying their mode of action. Further validation of the current findings will require appropriate xenograft models, which will also permit the assessment of peptide action within the context of PCa’s complex microenvironment [81]. Should the xenograft studies demonstrate encouraging results, the pharmacokinetics of peptides including processes of absorption, distribution, metabolism, and excretion will need to be assessed in future studies. Despite the poor stability in vivo reported for various bioactive peptides, extensive research is being conducted to address this issue, involving encapsulation technologies and chemical modifications. Compared with antibodies and other small therapeutics, peptides are, however, considered superior options in many regards [82,83]. With the exception of antimicrobial peptides, bioactive peptides are generally regarded as safe. For bioactive peptides to be considered therapeutics, however, they must be supported by an adequately high level of evidence supporting their safety.

## 4. Materials and Methods

### 4.1. Materials and Reagents

Human prostate cancer cell lines LNCaP (Cat. No. CRL-3313), PC3 (Cat. No. CRL-1435), and VCaP cells (Cat. No. CRL-2876) were obtained from the American Type Culture Collection (ATCC; Manassas, VA, USA); VCaP-EnzR (Cat. No. SCC421) cells were obtained from Millipore via Sigma-Aldrich (St. Louis, MO, USA). All cells tested negative for *Mycoplasma* contamination. Roswell Park Memorial Institute (RPMI) cell culture medium 1640 was purchased from Lonza Bioscience (Morrisville, NC, USA); L-glutamine, penicillin, and streptomycin was purchased from Thermo Fischer Scientific (Waltham, MA, USA); 10% heat-inactivated fetal bovine serum (FBS) was obtained from Sigma-Aldrich (Cat. No. ES-009-B) ≥ 98% bicalutamide powder (Cat. No. B9061), dimethyl sulfoxide (DMSO) and Dulbecco’s Modified Eagle’s Medium (DMEM) were obtained from ATCC (Cat. No. 30-2002); high-glucose DMEM (EMD Millipore, Cat. No. SLM-120-B) ≥ 99% enzalutamide powder was obtained from Intas Pharmaceuticals Ltd. (Ahmedabad, India). Eight custom peptides were synthesized as their HCl salts by Biomatik (Kitchener, ON, Canada). The primary structure of the peptides was as follows: FT-001 (Arg-Glu-Glu-Ser-Gly-Glu); FT-002 (Arg-Glu-Glu-Ser-Gly-Glu-Pro); FT-003 (Lys-Glu-Glu-Asp-Glu-Glu-Ser-Gly-Glu); FT-004 (Lys-Pro-Arg-Glu-Glu-Ser-Gly-Glu); FT-005 (Leu-Asp-Glu-Glu-Ser-Gly-Glu-Pro); FT-006 (Arg-Glu-Glu-Ser-Asp-Lys-Pro-Asn-Tyr); FT-007 (Pro-Arg-Glu-Glu-Ser-Asp-Lys-Pro), FT-008 (Arg-Glu-Glu-Ser-Gly-Glu-Leu). SPH was prepared via the enzymatic hydrolysis of salmon raw materials and was obtained from Hofseth Biocare ASA (Keiser Wilhelmsgate 24, Møre og Romsdal, 6003 Ålesund, Norway). SPH appears as a light-yellow powder. It has a water-soluble protein content of >95%, of which >25% is composed of type I/III collagen peptides, fat content < 0.5%, and ash content < 2.5%. Amino acid composition is glutamic acid (13.9 g/100 g), aspartic acid (9.4 g/100 g), glycine (14.9 g/100 g), proline (7.6 g/100 g), lysine (7.0 g/ 100 g), alanine (7.5 g/100 g), and arginine (6.9 g/100 g). Average molecular weight (MW) of peptides in SPH is determined to be 3395 Dalton.

### 4.2. Cell Culture Preparation

LNCaP and PC3 cells were maintained in RPMI 1640 medium enriched with 2 mmol/L L-glutamine, 100 U/mL penicillin, 100 mg/mL streptomycin, and 10% heat-inactivated FBS. The base for VCaP was DMEM with added FBS to a final concentration of 10%. High-glucose DMEM supplemented with 10% FBS was added to VCaP-EnzR cells. Cells were incubated in a 5% CO_2_-incubator at body temperature (37 °C) and high humidity.

### 4.3. Test Solutions

Testing solutions at desired concentrations were prepared by sequential dilution of a 1000 µM (0.001 M) standard solution in 3% DMSO/PBS solution. This protocol was applied to synthetic peptides and the SPH. Based on manufacturer-provided information regarding solubility, a stock solution of 10 µM was prepared for the peptides. Prior to use, sonication was 10 min was performed. Bicalutamide test solutions were prepared by serially diluting the substance in DMEM according to the manufacturer’s instructions. To achieve working concentrations, enzalutamide was diluted in DMSO in accordance with the manufacturer’s instructions.

### 4.4. Peptide Design and Synthesis

Based on a bioassay-guided fractionation procedure, the most bioactive fraction of an SPH was isolated with regard to its ability to modulate *FTH1* and *TFRC*. Based on the analysis of fragmentation patterns using LC/MS/MS, a common EESGE core peptide sequence was identified as being responsible for most of the gene modulatory properties. Custom peptide synthesis was carried out by Biomatik (Kitchener, ON, Canada). At the N- and C-termini, specific amino acid residues were added to improve bioavailability and cell penetration.

### 4.5. Clonogenic Assay

LNCaP, PC3, VCaP, and VCaP-EnzR were seeded in 10-mm welled petri dishes at a density of 3000–5000 cells per well in RPMI 1640 and incubated. After 24 h, cells were cultured with their respective compound(s) at indicated concentrations. Treatments were applied daily for five days without changing the media, and excess solution was discarded. On day 12, cells were fixed, washed, and stained with crystal violet and counted with a TC20 Automated Cell Counter (Bio-Rad, Hercules, CA, USA). Cells in regular cell medium and 0.1% DMSO were employed as internal controls. The anti-proliferative effects of peptides were compared using IC_50_ values of bicalutamide and enzalutamide for its respective cell lines. IC_50_ was extrapolated from the dose-response graph in Excel. Relative plating efficiencies were expressed as percentages relative to the plating efficiency of untreated cells and reported as a relative percent colony survival rate. Experiments were performed in triplicate.

### 4.6. Differential Gene Expression

Changes in mRNA expression levels of *FTH1* and *TFRC* genes between experimental groups and control groups were investigated. The four most potent peptides with respect to anti-proliferative effects were chosen for analysis. Subsequently, fold-change analysis was conducted. Total RNA was isolated from cells with a one-step liquid phase separation using uPzol™ RNA Isolation Solution (Biotechrabbit, Berlin, Germany). DNA contamination was cleared using TURBO dNase, as specified by the manufacturer. RNA to single-stranded cDNA was accomplished with a high-capacity Reverse Transcriptase Kit (Applied Biosystems; Waltham, MA, USA). A total of 1 µL cDNA (corresponding to 50 ng RNA) was amplified with RT-PCR (QuantStudio™ 6 Flex RT PCR System) in a TaqMan Assay (Thermo Fisher Scientific; Waltham, MA, USA) and TaqMan™ Universal PCR Master Mix (Catalogue No. 4304437). Wells containing DMSO 0.04% were used as the negative control. TaqMan probes used were *FTH1* (Hs01694011_s1), *TFRC* (Hs00951083_m1), and Housekeeping *ACTB* (Hs01060665_g1). Quantitative PCR data analysis was conducted with the 2^−ΔΔCT^ method with *ACTB* as the reference gene. Normalized expression after log2-transformation of the fold change was established.

### 4.7. Statistical Analysis

The Kruskal–Wallis test by ranks and Dunn’s test with Šidák correction for multiple pairwise comparisons were used to assess statistical significance in the relative colony formation potential between the treated and control groups. Kruskal–Wallis *p* < 0.05 concludes that groups have difference distributions. For Dunn’s test, differences were considered statistically significant when the *p*-value was lower than 0.05 (*) and 0.01 (**). Every experiment was performed in triplicate.

## Figures and Tables

**Figure 1 ijms-24-15231-f001:**
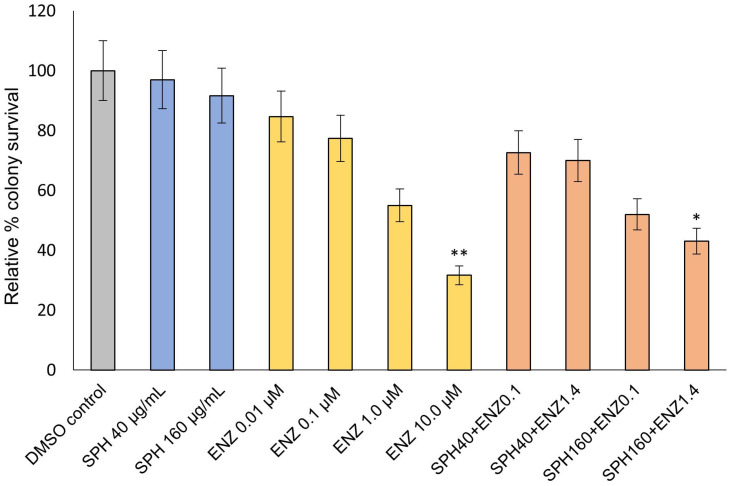
VCaP clonogenic potential following different treatments and doses. Data are presented as mean ± standard deviation (SD) of three independent experiments. * *p* < 0.05; ** *p* < 0.01.

**Figure 2 ijms-24-15231-f002:**
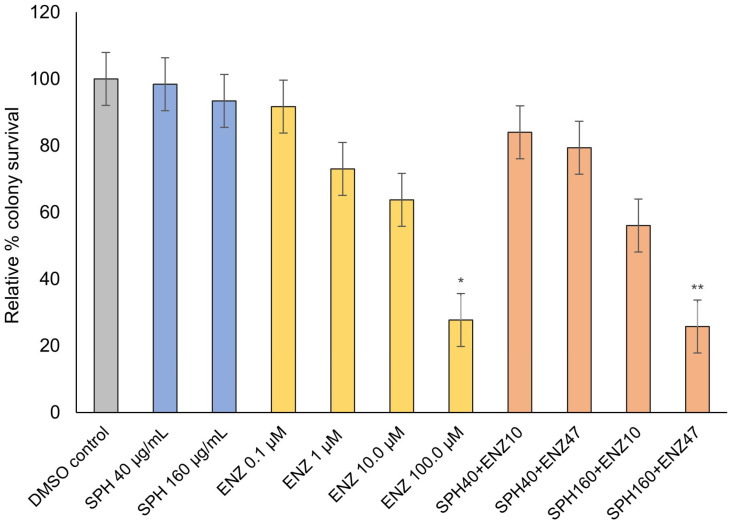
VCaP-EnzR clonogenic potential following different treatments and doses. Data are presented as mean ± SD of three independent experiments. * *p* < 0.05; ** *p* < 0.01.

**Figure 3 ijms-24-15231-f003:**
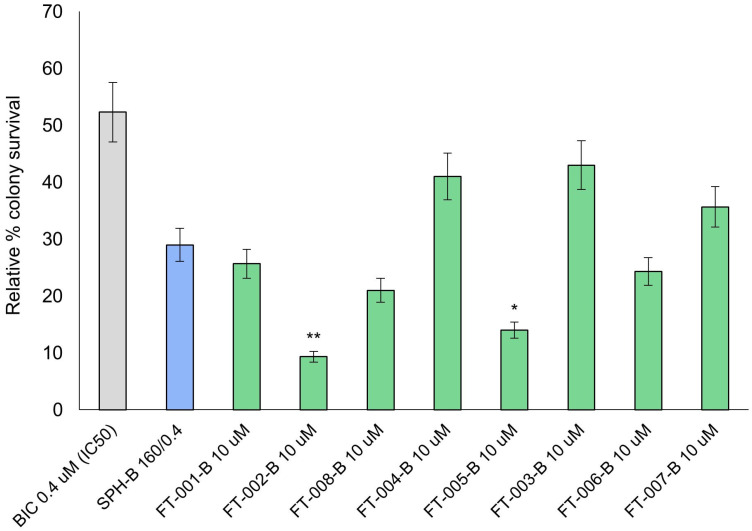
LNCaP clonogenic potential following different treatments and doses. Data are presented as mean ± SD of three independent experiments. SPH-B 160/0.4 is defined as 160 µg/mL soluble protein hydrolysate (SPH) with 0.4 µM (IC_50_) bicalutamide. FT-XXX-B denotes 10 uM peptide stock solution co-administered with 0.4 µM (IC_50_) bicalutamide; * *p* < 0.05; ** *p* < 0.01.

**Figure 4 ijms-24-15231-f004:**
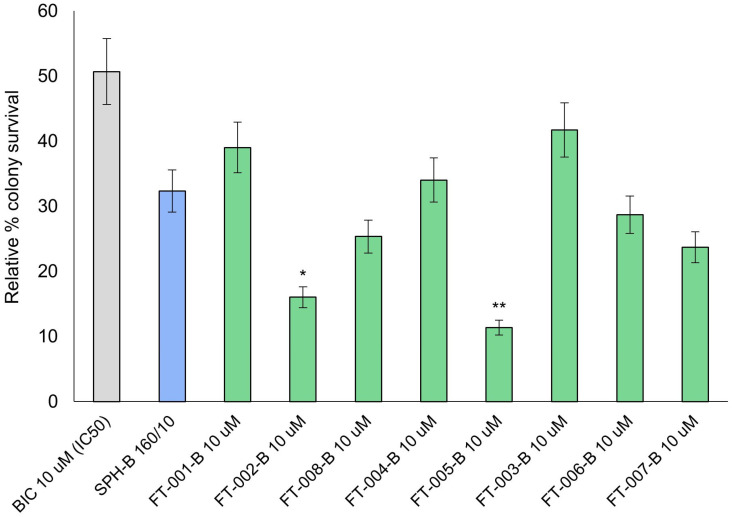
PC3 clonogenic potential following different treatments and doses. Data are presented as mean ± (SD) of three independent experiments. SPH-B 160/10 is defined as 160 µg/mL SPH with 10.0 µM (IC_50_) bicalutamide. FT-XXX-B denotes 10 uM peptide stock solution co-administered with 10 µM (IC_50_) bicalutamide; * *p* < 0.05; ** *p* < 0.01.

**Figure 5 ijms-24-15231-f005:**
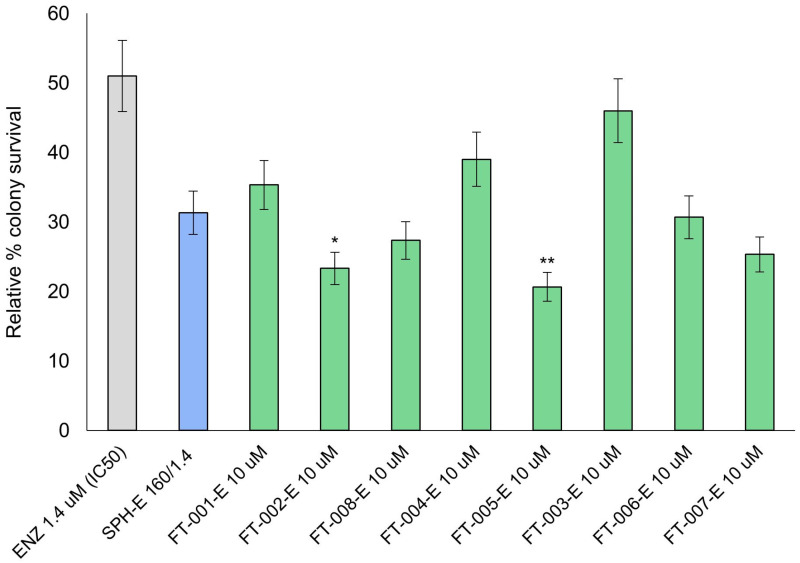
VCaP clonogenic potential following different treatments and doses. Data are presented as mean ± SD of three independent experiments. SPH-B 160/1.4 is defined as 160 µg/mL SPH with 1.4 µM (IC_50_) enzalutamide. FT-XXX-E denotes 10 µM peptide stock solution co-administered with 1.4 µM (IC_50_) enzalutamide; * *p* < 0.05; ** *p* < 0.01.

**Figure 6 ijms-24-15231-f006:**
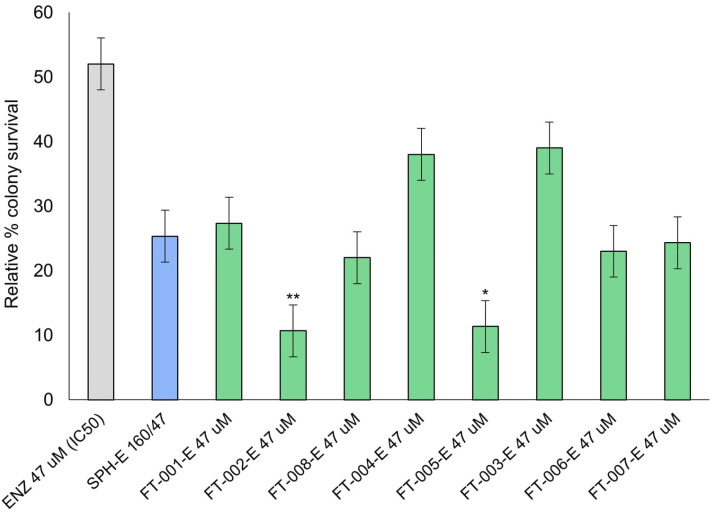
VCaP-EnzR clonogenic potential following different treatments and doses. Data are presented as mean ± SD of three independent experiments. SPH-B 160/47 is defined as 160 µg/mL SPH with 47 µM (IC_50_) enzalutamide. FT-XXX-E denotes 10 µM peptide stock solution co-administered with 47 µM (IC_50_) enzalutamide; * *p* < 0.05; ** *p* < 0.01.

**Figure 7 ijms-24-15231-f007:**
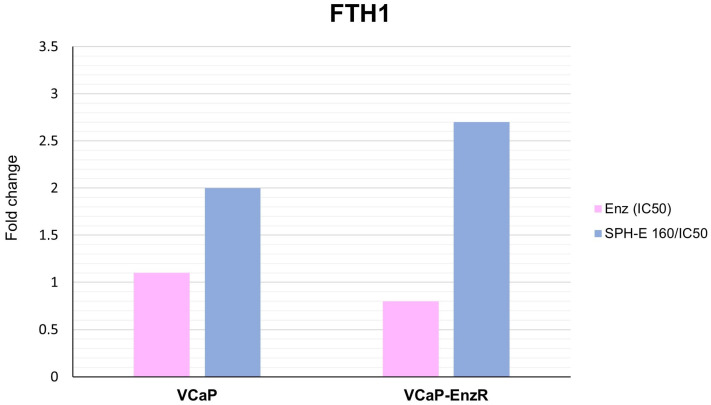
Fold change in *FTH1* mRNA expression in VCaP and VCaP-EnzR cells following treatments. mRNA levels were determined by quantitative real-time polymerase chain reaction (qRT-PCR) normalized to the internal control gene for beta actin (*ACTB*). Results are presented as relative values (fold change) determined by the 2^–∆∆Ct^ method. SPH-E 160/IC50 denotes SPH 160 µg/mL co-administered with respective IC_50_ enzalutamide concentrations.

**Figure 8 ijms-24-15231-f008:**
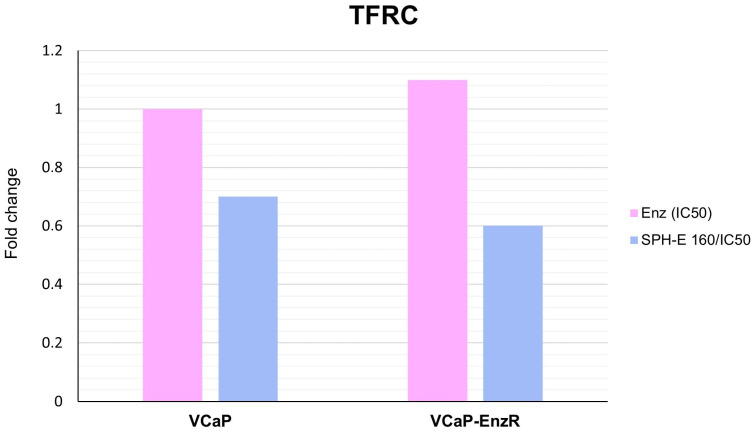
Fold change in *TFRC* mRNA expression in VCaP and VCaP-EnzR cells following treatments. SPH-E 160/IC50 denotes SPH 160 µg/mL co-administered with respective IC_50_ enzalutamide concentrations.

**Figure 9 ijms-24-15231-f009:**
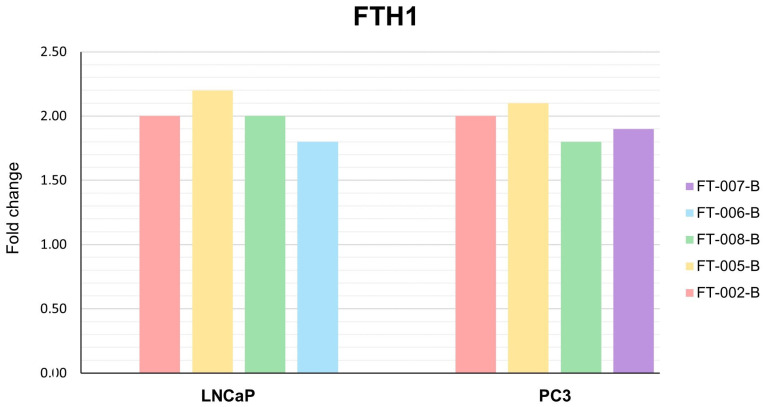
Fold change in *FTH1* mRNA expression in LNCaP and PC3 cells following different treatments. Data are presented as mean of three independent experiments. FT-XXX-B denotes 10 µM peptide stock solution co-administered with 0.4 µM (IC_50_) bicalutamide.

**Figure 10 ijms-24-15231-f010:**
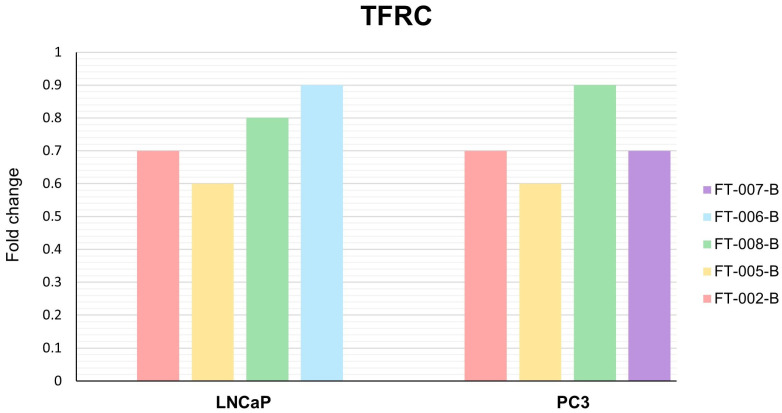
Fold change in *TFRC* mRNA expression in LNCaP and PC3 cells following different treatments. Data are presented as mean of three independent experiments. FT-XXX-B denotes 10 µM peptide stock solution co-administered with 0.4 µM (IC_50_) bicalutamide.

**Figure 11 ijms-24-15231-f011:**
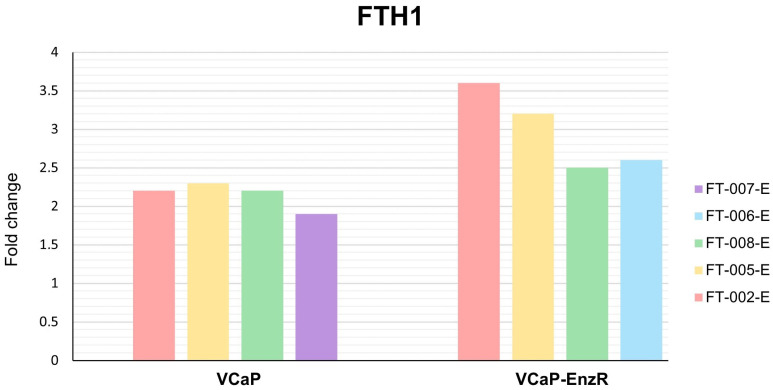
Fold change in *FTH1* expression in VCaP and VCaP-EnzR cells following different treatments. Data are mean of three independent experiments. FT-XXX-B denotes 10 µM peptide stock solution co-administered with 0.4 µM (IC_50_) bicalutamide.

**Figure 12 ijms-24-15231-f012:**
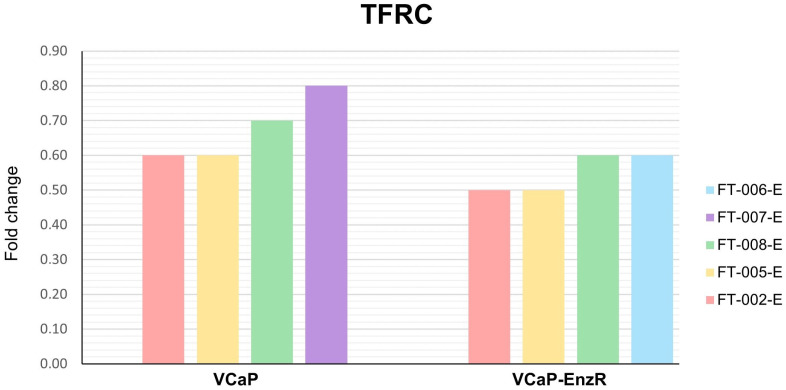
Fold change in *TFRC* expression in VCaP and VCaP-EnzR cells following different treatments. Data are mean of three independent experiments. FT-XXX-B denotes 10 µM peptide stock solution co-administered with 0.4 µM (IC_50_) bicalutamide.

## Data Availability

Data requests can be directed to the authors of this paper.

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
