# Peer review of "Assessing the Potential of Small Peptides for Altering Expression Levels of the Iron-Regulatory Genes FTH1 and TFRC and Enhancing Androgen Receptor Inhibitor Activity in In Vitro Prostate Cancer Models"

_ijms, 2023, doi:10.3390/ijms242015231_

Round 1
Reviewer 1 Report
The manuscript submitted by Currie and co-workers shows the modulatory effect of small peptides on FTH1 and TFRC Expression as well as enhanced activity of androgen receptor in prostate cancer cell model. The article is original, well structured; easy to read with main emphasis on the effect of eight small peptides against two targets FTH1 and TFRC four prostate cancer cell lines. In my opinion, the manuscript can be published in this journal, after the authors have addressed the following minor issues:
1. Abstract and Introduction sections are not written in a proper manner. Title of the manuscript is not properly designed and formatting is not there.
2. Only one cell line is selected for the study, there should be at least two cell lines for validation of study if in vivo study is not there.
3. Please clear the concept behind selection of specific doses. For better understanding authors must provide the data of these doses in normal bladder or any cells.
4. Authors must improve the graph and bar diagram presentation as well as resolution.
5. Authors should explain the results of each experiment in elaborated manner.
6. Very poor presentation of graph and diagram. The quality and resolution of each graph and diagram is not up to the mark. Stained cell diagram must be in proper resolution and large enough to clearly see the changes imparted by natural toxins.
7. Discussion section require more explanation and must be aligned with previous studies.
8. Conclusion should be improved with addition of future perspectives.
Moderate English proofreading is required
Reviewer 2 Report
This is a meaningful report with obvious translational value. It is suitable for publication, but I have some reservations over the following points:
* The “in an In Vitro Prostate Cancer Model” can be replaced with “in In Vitro Prostate Cancer Models” as it is clearly the case?
* In the abstract, the authors can consider combine the two sentences to clearly demonstrate the relationship between SPH and FTH1 and TFRC gene regulation: “Previously it was found that a soluble protein hydrolysate (SPH) enhanced the antiproliferative effects of an androgen receptor inhibitor (ARI) in prostate cancer cells. FTH1 and TFRC genes were modulated concurrently, implicating this mechanism in its antiproliferative action.”
* Is there a particular reason that the x axis of Fig. 3-6 has a layout of the synthesized peptide in an order of 001-002-008-004-005-003-006-007?
* Why were some peptides not tested in the FTH1 and TFRC expression level examinations?
* Does the lack of error bars in Fig. 7-10 indicate that the results were obtained with only one technical replicate? I trust the results are reproducible, but this may need to be clearly stated.
* Relating to the clinical potential of the developed peptides, can the authors comment on any clinical progress, and any in vivo experimental results in preclinical studies, related to iron dysmetabolism in prostate cancer treatment, either in Introduction or Discussion? While this manuscript focuses on in vitro validations, some background and perspectives related to in vivo and clinical applications can be beneficial.
* The authors may need to provide more details about how the specific peptide sequences were generated. It was not sufficiently discussed in the manuscript.
